# Arctan-Based Family of Distributions: Properties, Survival Regression, Bayesian Analysis and Applications

**Omid Kharazmi** [1], **Morad Alizadeh** [2], **Javier E. Contreras-Reyes** [3,*] and **Hossein Haghbin** [2]

1 Department of Statistics, Vali-e-Asr University of Rafsanjan, Rafsanjan 7718897111, Iran
2 Department of Statistics, Faculty of Intelligent Systems Engineering and Data Science, Persian Gulf University, Bushehr 7516913817, Iran
3 Facultad de Ciencias, Instituto de Estadística, Universidad de Valparaíso, Valparaíso 2360102, Chile
* Correspondence: jecontrr@uc.cl; Tel.: +56-(32)-250-8242

**Abstract:** In this paper, a new class of the continuous distributions is established via compounding the arctangent function with a generalized log-logistic class of distributions. Some structural properties of the suggested model such as distribution function, hazard function, quantile function, asymptotics and a useful expansion for the new class are given in a general setting. Two special cases of this new class are considered by employing Weibull and normal distributions as the parent distribution. Further, we derive a survival regression model based on a sub-model with Weibull parent distribution and then estimate the parameters of the proposed regression model making use of Bayesian and frequentist approaches. We consider seven loss functions, namely the squared error, modified squared error, weighted squared error, K-loss, linear exponential, general entropy, and precautionary loss functions for Bayesian discussion. Bayesian numerical results include a Bayes estimator, associated posterior risk, credible and highest posterior density intervals are provided. In order to explore the consistency property of the maximum likelihood estimators, a simulation study is presented via Monte Carlo procedure. The parameters of two sub-models are estimated with maximum likelihood and the usefulness of these sub-models and a proposed survival regression model is examined by means of three real datasets.

**Keywords:** arctangent function; bayesian estimation; maximum likelihood; loss function; odd log-logistic distribution; survival regression; statistical distribution

**MSC:** 60E05; 62H10



## 1. Introduction

Distribution theory provides useful tools in describing and identifying the model of occurred events and predicting next events. Recently, several generators of probability distributions have been introduced by many researchers in the statistical literature. Some well-known generators are the Marshall–Olkin generated (MO-G) by [1], beta-G by [2], Kumaraswamy-G (Kw-G) by [3], Weibull-G by [4], exponentiated half-logistic-G by [5], Lomax-G by [6], and polar-generalized normal distribution by [7], among others.

A favorite technique in expanding statistical distributions is the method introduced by [8], who have introduced the generalized log-logistic (GLL-G) class of distributions. The cumulative distribution (cdf) function of this class based on underline cdf $G$, is given by

$$\Pi(x;\beta) = \left[ G(x)^{\beta} + \bar{G}(x)^{\beta} \right]^{-1} \times G(x)^{\beta}, \tag{1}$$

where $\beta > 0$ and $\bar{G}(x) = 1 - G(x)$ denote the survival function. This class has named by Odd log-logistc (OLL-G) and several extensions of this class were introduced. Kumaraswamy Odd log-logistic due to [9], beta odd log-logistic due to [10], odd burr general-

ized class due to [11], Topp–Leone odd log-logistic due to [12], generalized odd log-logistic due to [13], new odd log-logistic due to [14] and odd log-logistic logarithmic by [15].

The purposes of this work are two fold. We first introduce a general and versatile class of distributions in terms of compounding the arctan function and cdf defined in (1). This model is referred to as the arctan odd log-logistic-G (ATOLL-G) distribution. The second purpose of this work lies in the study of two sub-models of the general ATOLL-G model via classical and Bayesian approaches. Further, we study the corresponding regression model derived from sub-model which is defined based on the Weibull distribution. First, certain statistical and reliability properties of the ATOLL-G distribution are derived in a general setting. Then, we establish two special cases of ATOLL-G by using the Weibull and normal distributions instead of the parent distribution *G*. These models are called ATOLL-W and ATOLL-N distribution, respectively. We also provide a discussion for the ATOLL-W regression model via log-transformation of ATOLL-W (LATOLL-W) distribution. Furthermore, we obtain Bayesian and maximum likelihood estimates of the parameters of proposed models via real examples.

For Bayesian inference, we consider several asymmetric and symmetric loss functions such as squared error loss, modified squared error, precautionary, weighted squared error, linear exponential, general entropy, and *K*-loss functions to estimate the parameters of the LATOLL-W regression model. Further, making use of the independent prior distributions, Bayesian 95% credible and highest posterior density (HPD) intervals (see [16]) are provided for each parameter of the proposed model. In addition, a simulation study is performed to investigate Maximum Likelihood Estimators (MLEs) of consistency.

The rest of the manuscript is organized as follows. In Section 2, we introduce a new class of distributions called arctan odd log-logistic-G (ATOLL-G) distribution. Some structural properties of the ATOLL-G distribution such as the hazard function, quantiles, asymptotics and some useful expansions of the proposed model are given in a general setting in Section 3. In Section 4, two special cases of this class is considered by employing Weibull and normal distributions as the parent distribution. The ATOLLW regression model and its Bayesian inference are presented by considering seven well-known loss functions in Section 5. In Section 6, we study the performance of the maximum likelihood estimates of the parameters of ATOLLW distribution via Monte Carlo simulation to investigate the mean square error and bias of the maximum likelihood estimators. In Section 7, the supremacy of the ATOLLN and ATOLLW models to some challenger models is exhibited via several selection model criteria by analyzing Data 1 and Data 2 real examples, respectively. Further, we fit the LATOLLW regression model to heart transparent dataset and compare its efficiency with some competitor models. We also provide the numerical results of Bayesian inference and related plots to posterior samples for heart transplant data in this Section. Finally, the paper is concluded in Section 8.

## 2. Model Genesis

In this section, we first introduce an unit-interval distribution based on arctan function. Then we propose arctan odd log logistic *G* class of distributions.

### 2.1. A New Extension of Uniform Distribution in Terms of Arctan Function

We create a new unit-interval distribution, based on the definition of arctan function with closed-form cdf given by:

$$F(x) = \frac{\arctan(\alpha\,x)}{\arctan(\alpha)},\ 0 < x < 1,\ \alpha > 0. \tag{2}$$

The related probability density function (pdf) is obtained by:

$$f(x) = \frac{\alpha}{\arctan(\alpha)(1 + \alpha^2\,x^2)},\ 0 < x < 1,\ \alpha > 0. \tag{3}$$

To study the effect of $\alpha$ on the pdf in (3), we plot this pdf under some selected values of the parameter $\alpha$ in Figure 1.

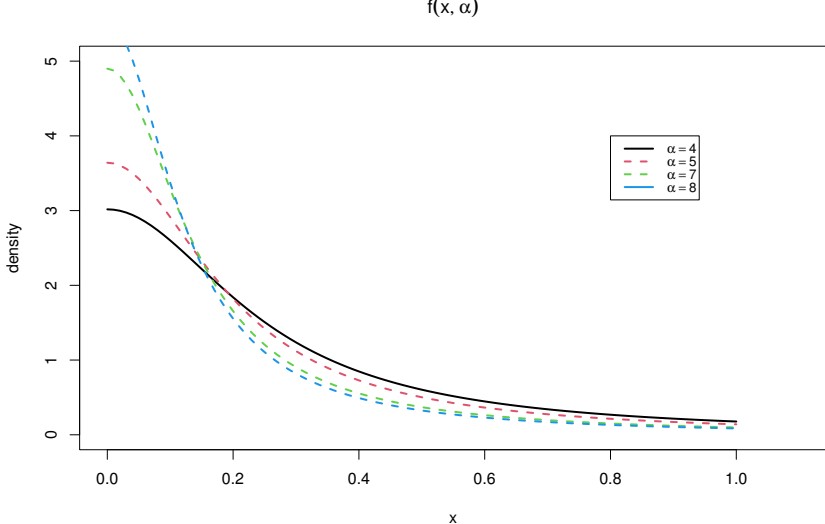

**Figure 1.** Plots of the pdf in (3).

It is worthwhile to note that when $\alpha \to 0^+$, the pdf in (3) reduces to standard uniform distribution.

The first four moments of the pdf in (3) are given by:

$$E(X) = \frac{\log(\alpha^2 + 1)}{2\alpha \, \arctan(\alpha)};$$

$$E(X^2) = \frac{\alpha - \arctan(\alpha)}{\alpha^2 \, \arctan(\alpha)};$$

$$E(X^3) = \frac{\alpha^2 - \log(\alpha^2 + 1)}{2 \, \alpha^3 \, \arctan(\alpha)};$$

$$E(X^4) = \frac{\alpha(\alpha^2 - 3) + 3 \, \arctan(\alpha)}{3 \, \alpha^4 \, \arctan(\alpha)}.$$

### 2.2. Arctan Odd Log Logistic G Family of Distributions

Here, we propose a general class of distributions in terms of compounding the arctan function and cdf in (1). The cdf of arctan odd log logistic $G$ class of continuous distributions is given as:

$$F(x; \alpha, \beta) = \frac{\arctan(\alpha \, \Pi(x; \beta))}{\arctan(\alpha)}, \tag{4}$$

where $\Pi(x; \beta)$ is defined as in (1). This model is called arctan odd log logistic G (ATOLL-G) distribution. The corresponding pdf is also given by:

$$f(x; \alpha, \beta) = \frac{\alpha \, \pi(x; \beta)}{\arctan(\alpha)[1 + (\alpha \, \Pi(x; \beta))^2]}, \tag{5}$$

where $\pi(x)$ is defined by

$$\pi(x) = \beta \, g(x; \beta) G(x)^{\beta - 1} \, \bar{G}(x)^{\beta - 1} \left[ G(x)^\beta + \bar{G}(x)^\beta \right]^{-2}. \tag{6}$$

Note that when $\alpha \to 0^+$, the pdf in (5) reduces to OLL-G family. Further, when $\alpha \to 0^+$ and $\beta = 1$, it reduces to baseline distribution $G$. We can readily obtain the associated hazard rate function of (4) as:

$$\psi(x) = \frac{\alpha\,\pi(x)}{\{\arctan(\alpha) - \arctan(\alpha\,\Pi(x))\}[1 + (\alpha\,\Pi(x))^2]}. \tag{7}$$

## 3. Properties

In this section we study some basic properties of the ATOLL-G family.

### 3.1. Quantile Function

If $U \sim U(0,1)$, and $Q_G(u) = G^{-1}(u)$ denote the quantile function of $G$, then:

$$X_U = Q_G \left\{ \frac{[\tan(U\,\arctan(\alpha))]^{\frac{1}{\beta}}}{[\tan(U\,\arctan(\alpha))]^{\frac{1}{\beta}} + [\alpha - \tan(U\,\arctan(\alpha))]^{\frac{1}{\beta}}} \right\} \tag{8}$$

has its cdf as in (4).

### 3.2. Asymptotics

Suppose $X$ be a positive random variable. Then, the asymptotics of Equations (4), (5) and (7) as $x \to 0^+$ are given by:

$$F(x) \sim \frac{\alpha\,G(x)^\beta}{\arctan(\alpha)},$$

$$f(x) \sim \frac{\alpha\,\beta\,g(x)\,G(x)^{\beta-1}}{\arctan(\alpha)},$$

$$\psi(x) \sim \frac{\alpha\,\beta\,g(x)\,G(x)^{\beta-1}}{\arctan(\alpha) - \alpha\,G(x)^\beta}.$$

The asymptotics of Equations (4), (5) and (7) as $x \to \infty$ are given by:

$$1 - F(x) \sim \frac{\alpha\beta\,\bar{G}(x)}{\arctan(\alpha)},$$

$$f(x) \sim \frac{\alpha\beta\,g(x)}{\arctan(\alpha)},$$

$$\psi(x) \sim \frac{g(x)}{\bar{G}(x)}.$$

### 3.3. Probability Density and Cumulative Density Function Expansion Series

For a given cdf $G(x)$, a variable $Z$ has the exp-G distribution with power parameter $\eta > 0$, say $Z \sim$ exp-G$(\eta)$, if the related pdf and cdf are given by:

$$h_\eta(x) = \eta\,G(x)^{\eta-1}\,g(x) \quad \text{and} \quad H_\eta(x) = G(x)^\eta,$$

respectively. For pertinent details, one can see [17–19].

First note that we can write:

$$\arctan(x) = \sum_{l=0}^{\infty} \frac{(-1)^l\,x^{2l+1}}{2l+1}, \tag{9}$$

For more details, see [20]. The cdf in (4) can also be represented as:

$$F(x) = \frac{1}{\arctan(\alpha)} \sum_{l=0}^{\infty} \frac{(-1)^l\,\alpha^{2l+1}G(x)^{\beta\,(2l+1)}}{(2l+1)\left[G(x)^\beta + \bar{G}(x)^\beta\right]^{2l+1}}. \tag{10}$$

Since

$$\left[ G(x)^\beta + \bar{G}(x)^\beta \right]^{-2l-1} = \sum_{i=0}^\infty \binom{-2l-1}{i} \left\{ G(x)^\beta - \left[ 1 - \bar{G}(x)^\beta \right] \right\}^i$$

$$= \sum_{i=0}^\infty \sum_{j=0}^i (-1)^{i-j} \binom{-2l-1}{i} \binom{i}{j} G(x)^{\beta j} \left[ 1 - \bar{G}(x)^\beta \right]^{i-j}$$

$$= \sum_{j=0}^\infty \sum_{i=j}^i (-1)^{i-j} \binom{-2l-1}{i} \binom{i}{j} G(x)^{\beta j} \left[ 1 - \bar{G}(x)^\beta \right]^{i-j}$$

$$= \sum_{j,r=0}^\infty \sum_{i=j}^i \sum_{k=0}^{i-j} (-1)^{i-j+k+r} \binom{-2l-1}{i} \binom{i}{j} \binom{i-j}{k} \binom{\beta k}{r} G(x)^{\beta j+r},;$$

then, we obtain:

$$F(x) = \sum_{j,l,r=0}^\infty w_{l,j,r}\, G(x)^{\beta\,(2l+j+1)+r} = \sum_{j,l,r=0}^\infty w_{l,j,r}\, H_{\beta\,(2l+j+1)+r}(x),$$

where

$$w_{l,j,r} = \sum_{i=j}^i \sum_{k=0}^{i-j} \frac{\alpha^{2l+1}(-1)^{l+i-j+k+r} \binom{-2l-1}{i} \binom{i}{j} \binom{i-j}{k} \binom{\beta k}{r} G(x)^{\beta j+r}}{(2l+1)\arctan(\alpha)}.$$

Therefore, the density function of $X$ can be obtained as an infinite linear combination of exp-G density functions

$$f(x;\alpha,\beta,,\xi) = \sum_{j,l,r=0}^\infty w_{l,j,r}\, h_{\beta\,(2l+j+1)+r}(x), \tag{11}$$

where $h_{\beta\,(2l+j+1)+r}(x;\xi) = (\beta\,(2l+j+1)+r)\, g(x;\xi)\, G(x;\xi)^{\beta\,(2l+j+1)+r-1}$ represent the exp-G density function with power parameter $\beta\,(2l+j+1)+r$. Hereafter, a random variable with the density function $h_{k+1}(x;\xi)$ is denoted by $Y_{k+1} \sim$ exp-G$(k+1)$. Equation (11) reveals that the ATOLL-G density function is a linear combination of exp-G densities. We can obtain some mathematical properties of the ATOLL-G based on EXP-G densities, for example we can obtain moments, incomplete moments, moment generating function and linear combination for order statistics.

## 4. Two Sub-Models

In this section, we propose two special cases of ATOLLG distribution, which are used in squeal.

### 4.1. Arctan Odd Log-Logistic Weibull Distribution

Suppose that the parent distribution $G$ has Weibull distribution with cdf $G(x) = 1 - e^{-(\lambda x)^\gamma}$, then from (5), the pdf of arctan odd log-logistic Weibull distribution (ATOLLW) is defined by:

$$f(x) = \frac{\alpha}{\arctan(\alpha)} \frac{\beta\gamma\lambda^\gamma x^{\gamma-1} e^{-\beta(\lambda x)^\gamma} \left( 1 - e^{-(\lambda x)^\gamma} \right)^{\beta-1}}{\left\{ (1-e^{-(\lambda x)^\gamma})^\beta + e^{-\beta(\lambda x)^\gamma} \right\}^2 + \alpha^2 \left( 1 - e^{-(\lambda x)^\gamma} \right)^{2\beta}}, \quad x > 0. \tag{12}$$

From (4), the corresponding cdf is given by:

$$F(x) = \frac{1}{\arctan(\alpha)} \arctan\left\{\alpha \frac{\left(1 - e^{-(\lambda x)^{\gamma}}\right)^{\beta}}{\left(1 - e^{-(\lambda x)^{\gamma}}\right)^{\beta} + e^{-\beta(\lambda x)^{\gamma}}}\right\}. \quad (13)$$

The density of ATOLLW distribution under some selected values of associated parameters is plotted in Figure 2.

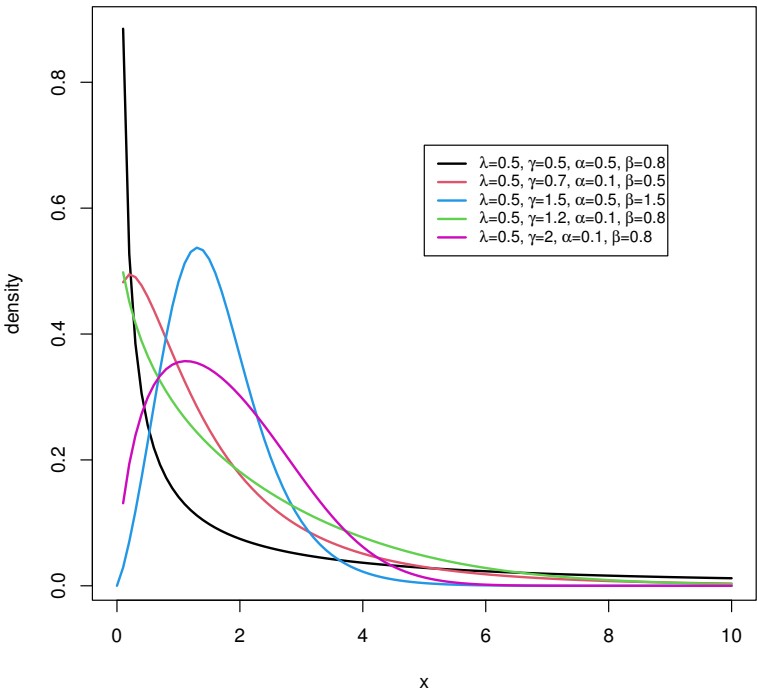

**Figure 2.** Plots of the pdf in (12).

*4.2. Arctan Odd Log-Logistic Normal Distribution*

Let the parent distribution $G$ have normal distribution with cdf $\Phi(x; \mu, \sigma^2)$. From (1), the pdf of arctan odd log-logistic normal distribution (ATOLLN) is defined by:

$$f(x) = \frac{\alpha}{\arctan(\alpha)} \frac{\beta \phi(x; \mu, \sigma^2) \Phi(x; \mu, \sigma^2)^{\beta-1} \bar{\Phi}(x; \mu, \sigma^2)^{\beta-1}}{\left[\Phi(x; \mu, \sigma^2)^{\beta} + \bar{\Phi}(x; \mu, \sigma^2)^{\beta}\right]^2 + \left(\alpha \Phi(x; \mu, \sigma^2)^{\beta}\right)^2}, \quad -\infty < x < \infty, \quad (14)$$

where $\phi(x; \mu, \sigma^2)$ is the pdf of a normal distribution with mean $\mu$ and variance $\sigma^2$. From (4), the corresponding cdf of (14), is given by:

$$F(x) = \frac{1}{\arctan(\alpha)} \arctan\left(\alpha \frac{\Phi(x; \mu, \sigma^2)^{\beta}}{\Phi(x; \mu, \sigma^2)^{\beta} + \bar{\Phi}(x; \mu, \sigma^2)^{\beta}}\right). \quad (15)$$

We plot the density of ATOLLN distribution under some selected values of associated parameters in Figure 3.

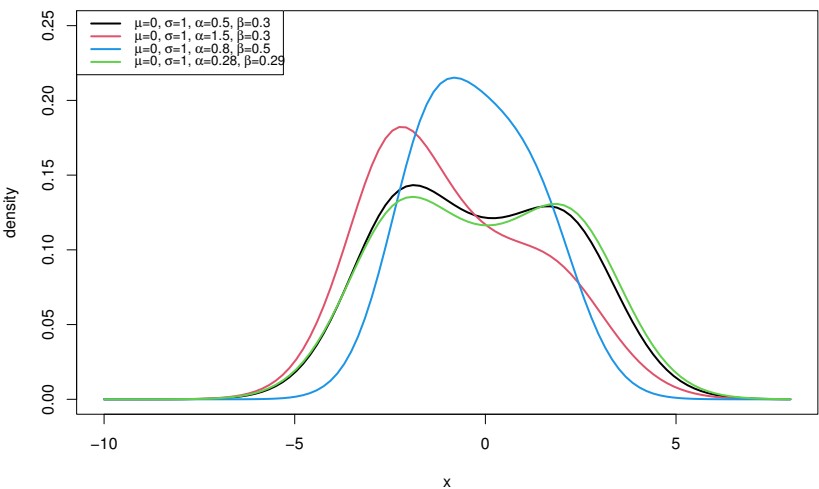

**Figure 3.** Plots of the pdf in (14).

## 5. The ATOLLW Regression Model

The survival regression model is one of well-known models in survival analysis. Sometimes for analyzing a lifetime variable, there are auxiliary information (as independent variables) that help us to explore the lifetime variable more precisely. More recently, by considering the class of location statistical distributions, different regression models have been introduced in the applied statistical literature (for example see [13,21]). The log-odd log-logistic Weibull regression model for censored data was introduced by [22] in terms of odd log-logistic Weibull distribution. Further, Cordeiro et al. [23] introduced a general regression model based on the Burr XII system of densities and also the log-odd power Cauchy–Weibull regression proposed by [24].

Let $X$ be a variable with pdf ATOLL-W defined in (12). Making use of the log transformation $Y = \ln(X)$, the pdf of transformed variable $Y$ is given by:

$$f_Y(y) = \frac{\alpha\beta}{\sigma\arctan(\alpha)} \frac{e^{\frac{y-\mu}{\sigma}} e^{-\beta e^{\frac{y-\mu}{\sigma}}} \left(1 - e^{-e^{\frac{y-\mu}{\sigma}}}\right)^{\beta-1}}{\left\{(1 - e^{-e^{\frac{y-\mu}{\sigma}}})^\beta + e^{-\beta e^{\frac{y-\mu}{\sigma}}}\right\}^2 + \alpha^2 \left(1 - e^{-e^{\frac{y-\mu}{\sigma}}}\right)^{2\beta}}, \quad y \in \mathbb{R}, \quad (16)$$

where $\sigma > 0$ is a scale, $\beta > 0$ is a shape and $\mu \in \mathbb{R}$ is a location parameter. The model in (16) is referred to as log-ATOLL-W (LATOLL-W) distribution, and it is briefly shown by $Y \sim \text{LATOLL-W}(\beta, \sigma, \mu)$. The survival function of $Y$ is:

$$S(y) = 1 - \frac{1}{\arctan(\alpha)} \arctan\left\{\alpha \frac{(1 - e^{-e^{\frac{y-\mu}{\sigma}}})^\beta}{(1 - e^{-e^{\frac{y-\mu}{\sigma}}})^\beta + e^{-\beta e^{\frac{y-\mu}{\sigma}}}}\right\}. \quad (17)$$

Let $Z = (Y - \mu)/\sigma$ be the standardized random variable having pdf,

$$h(z; \alpha, \sigma) = \frac{\alpha\beta}{\arctan(\alpha)} \frac{e^z e^{-\beta e^z}(1 - e^{-e^z})^{\beta-1}}{\{(1 - e^{-e^z})^\beta + e^{-\beta e^z}\}^2 + \alpha^2(1 - e^{-e^z})^{2\beta}}, \quad z \in \mathbb{R}.$$

The ATOLL-W regression is defined by:

$$y_i = \mu + \sigma z_i = \mathbf{v}_i^\mathsf{T}\boldsymbol{\tau} + \sigma z_i, \quad i = 1, \ldots, n, \quad (18)$$

where $\boldsymbol{\tau} = (\tau_1, \cdots, \tau_p)^\mathsf{T}$ is parameter vector of regression model, $\mathbf{v}_i^\mathsf{T}$ is covariate variable vector and $z_i$ is an error of regression model with density $h(z; \alpha, \sigma)$. Further, under assumptions $y_i = \min\{\log(c_i), \log(x_i)\}$, where $\log(c_i)$ denotes log-censoring and $\log(x_i)$ follows (16), and represent the log-lifetime. Let $r$ is the number of uncensored observations,

then the log-likelihood function for $\boldsymbol{\psi} = (\beta, \sigma, \boldsymbol{\tau}^{\mathsf{T}})^{\mathsf{T}}$ in terms of sets $F$ (set of individuals with log-lifetime) and $C$ (set of individuals with log-censoring) is given by:

$$\ell(\boldsymbol{\psi}) = r\left\{ \log \alpha - \log \sigma + \log \beta \right\} + \sum_{i \in F} \left\{ \log u_i - \beta u_i + (\beta - 1) \log(1 - \mathrm{e}^{-u_i}) \right\} \quad (19)$$

$$+ \sum_{i \in F} \log\left\{ \left( (1 - \mathrm{e}^{-u_i})^{\beta} + \mathrm{e}^{-\beta u_i} \right)^2 + \alpha^2 (1 - \mathrm{e}^{-u_i})^{2\beta} \right\}$$

$$+ \sum_{i \in C} \log\left\{ 1 - \frac{1}{\arctan(\alpha)} \arctan\left( \frac{\alpha(1 - \mathrm{e}^{-u_i})^{\beta}}{(1 - \mathrm{e}^{-u_i})^{\beta} + \mathrm{e}^{-\beta u_i}} \right) \right\}, \quad (20)$$

where $u_i = e^{z_i}$, $z_i = (y_i - \mathbf{v}_i^{\mathsf{T}} \boldsymbol{\tau})/\sigma$. For example, we can use the `optim` function of R software to obtain the MLE of $\boldsymbol{\psi}$ by maximizing (19).

*5.1. Residual*

The martingale and modified deviance residuals (mdr) for the LATOLL-W regression are given respectively by:

$$r_{M_i} = \begin{cases} 1 + \ln\left\{ 1 - \frac{1}{\arctan(\alpha)} \arctan\left( \frac{\alpha(1 - \mathrm{e}^{-u_i})^{\beta}}{(1 - \mathrm{e}^{-u_i})^{\beta} + \mathrm{e}^{-\beta u_i}} \right) \right\}, & i \in F, \\ \ln\left\{ 1 - \frac{1}{\arctan(\alpha)} \arctan\left( \frac{\alpha(1 - \mathrm{e}^{-u_i})^{\beta}}{(1 - \mathrm{e}^{-u_i})^{\beta} + \mathrm{e}^{-\beta u_i}} \right) \right\}, & i \in C, \end{cases}$$

where $u_i = \mathrm{e}^{\frac{y_i - \mathbf{x}_i^{\mathsf{T}} \boldsymbol{\tau}}{\sigma}}$, and

$$r_{D_i} = \begin{cases} \operatorname{sign}(r_{M_i}) \left\{ -2 \left[ r_{M_i} + \log(1 - r_{M_i}) \right] \right\}^{1/2}, & \text{if } i \in F, \\ \operatorname{sign}(r_{M_i}) \left\{ -2 r_{M_i} \right\}^{1/2}, & \text{if } i \in C. \end{cases} \quad (21)$$

When the regression model is well-fitted to a given data, the mdr are normally distributed with zero men and unit variance.

*5.2. Bayesian Inference of Regression Model*

In this section, we consider the Bayesian inference of the parameters for the survival regression model, which is discussed in Section 5. Let the parameters $\alpha$, $\beta$ and $\sigma$ of the LATOLLW distribution have independent prior distributions as:

$$\alpha \sim Gamma(a, b), \beta \sim Gamma(c, d), \sigma \sim Gamma(e, f), \tau_i \sim N(\mu_i, \sigma_i^2), \quad i = 0, 1, 2, 3,$$

where $a$, $b$, $c$, $d$, $e$ and $f$ are positive and $\tau_i \in \mathbb{R}$, $i = 0, 1, 2, 3$. Under these assumptions, the joint prior density function is formulated as follows:

$$\pi(\alpha, \beta, \sigma, \underline{\tau}) = \prod_{i=0}^{3} \frac{1}{\sqrt{2\pi\sigma_i^2}} e^{\frac{(\tau_i - \mu_i)^2}{\sigma_i^2}} \frac{b^a d^c f^e}{\Gamma(a)\Gamma(c)\Gamma(e)} \alpha^{a-1} \beta^{c-1} \sigma^{e-1} e^{-(b\alpha + d\beta + f\sigma)}, \quad (22)$$

where $\underline{\tau} = (\tau_0, \tau_1, \tau_2, \tau_3)$.

Here, we consider several asymmetric and symmetric loss functions including: squared error loss function (SELF), modified squared error loss function (MSELF), weighted squared error loss function (WSELF), *K*-loss function (KLF), linear exponential loss function (LINEXLF), precautionary loss function (PLF) and general entropy loss function (GELF). For more details, see [25] and the references therein. In Table 1, we provide a summary of these loss functions and associated Bayesian estimators and posterior risks.

**Table 1.** Seven loss functions with Bayes estimator and related posterior risk.

| Loss Function | Bayes Estimator | Posterior Risk |
|---|---|---|
| $L_1 = SELF = (\theta - d)^2$ | $E(\theta|\underline{x})$ | $Var(\theta|\underline{x})$ |
| $L_2 = WSELF = \frac{(\theta-d)^2}{\theta}$ | $(E(\theta^{-1}|\underline{x}))^{-1}$ | $E(\theta|x) - (E(\theta^{-1}|x))^{-1}$ |
| $L_3 = MSELF = \left(1 - \frac{d}{\theta}\right)^2$ | $\frac{E(\theta^{-1}|x)}{E(\theta^{-2}|x)}$ | $1 - \frac{E(\theta^{-1}|x)^2}{E(\theta^{-2}|x)}$ |
| $L_4 = PLF = \frac{(\theta-d)^2}{d}$ | $\sqrt{E(\theta^2|\underline{x})}$ | $2\left(\sqrt{E(\theta^2|\underline{x})} - E(\theta|\underline{x})\right)$ |
| $L_5 = KLF = (\sqrt{\frac{d}{\theta}} - \sqrt{\frac{\theta}{d}})$ | $\sqrt{\frac{E(\theta|\underline{x})}{E(\theta^{-1}|\underline{x})}}$ | $2\left(\sqrt{E(\theta|\underline{x})E(\theta^{-1}|\underline{x})} - 1\right)$ |
| $L_6 = LINEXLF = e^{c(\theta-d)} - c(\theta - d) - 1$ | $-\frac{\log E\left(e^{-c\theta}|\underline{x}\right)}{c}$ | $\log E(e^{-c\theta}|\underline{x}) + cE(\theta|\underline{x})$ |
| $L_7 = GELF = (\frac{\theta}{d})^c - c\log(\frac{\theta}{d}) - 1$ | $\left(E(\theta^{-c}|\underline{x})\right)^{-\frac{1}{c}}$ | $cE(\log \theta|\underline{x}) + \log[E(\theta^{-c}|\underline{x})]$ |

For more details see [26]. Let $\varphi$ be a function defined as:

$$\varphi(\alpha, \beta, \sigma, \underline{\tau}) = \frac{\alpha^{n+a-1}\beta^{n+c-1}\sigma^{e-(1+n)}}{\arctan^n(\alpha)} e^{-(b\alpha+d\beta+f\sigma)+\sum_{i=0}^{3}\frac{\tau_i^2-2\tau_i\mu_i}{\sigma_i^2}} \, , \; \alpha > 0, \; \beta > 0, \; \sigma > 0.$$

Since the joint posterior distribution $\pi(\alpha, \beta, \sigma, \underline{\tau})$ is formulated as:

$$\pi^*(\alpha, \beta, \sigma, \underline{\tau}|data) \propto \pi(\alpha, \beta, \sigma, \underline{\tau})L(data). \tag{23}$$

Therefore, the joint posterior density is given by:

$$\pi^*(\alpha, \beta, \sigma, \underline{\tau}|\underline{x}, \mathbf{V}) = K\varphi(\alpha, \beta, \sigma, \underline{\tau})\prod_{i=1}^{n} \frac{e^{\frac{y_i-\mu}{\sigma}}e^{-\beta e^{\frac{y_i-\mu}{\sigma}}}\left(1 - e^{-e^{\frac{y_i-\mu}{\sigma}}}\right)^{\beta-1}}{\left\{(1 - e^{-e^{\frac{y_i-\mu}{\sigma}}})^\beta + e^{-\beta e^{\frac{y_i-\mu}{\sigma}}}\right\}^2 + \alpha^2\left(1 - e^{-e^{\frac{y_i-\mu}{\sigma}}}\right)^{2\beta}}, \tag{24}$$

where $\mathbf{V} = (\mathbf{v_1}, \dots, \mathbf{v_n})^\top$ is a known matrix that contains the auxiliary variables, $\mu_i = \mathbf{v_i}^\top \underline{\tau}$ and $K$ is given as:

$$K^{-1} = \int_R \int_0^\infty \int_0^\infty \int_0^\infty \varphi(\alpha, \gamma, \sigma)\prod_{i=1}^{n} \frac{e^{\frac{y_i-\mu}{\sigma}}e^{-\beta e^{\frac{y_i-\mu}{\sigma}}}\left(1 - e^{-e^{\frac{y_i-\mu}{\sigma}}}\right)^{\beta-1}}{\left\{(1 - e^{-e^{\frac{y_i-\mu}{\sigma}}})^\beta + e^{-\beta e^{\frac{y_i-\mu}{\sigma}}}\right\}^2 + \alpha^2\left(1 - e^{-e^{\frac{y_i-\mu}{\sigma}}}\right)^{2\beta}}d\alpha d\beta d\sigma d\underline{\tau},$$

where the outer integration stands for parameter vector $\underline{\tau}$.

## 6. Simulation

Here, we examine the performance of the maximum likelihood estimates associated to the ATOLLN$(\mu, \sigma, a, b)$ distribution in (14) with respect to sample size $n$. The simulation study is performed via the Monte Carlo procedure as follows:

1. Generate 5000 samples of size $n$ for the ATOLLN$(\mu, \sigma, a, b)$ distribution by using the relation (8);

2. Compute the maximum likelihood estimates of parameter vector $\theta = (\alpha, \beta, \mu, \sigma)$ for the one thousand samples, say $\widehat{\theta}_{ij}$, for $i = 1, 2, \dots, 5000$; $j = 1, 2, 3, 4$;

3. Compute diagonal elements of inverse Fisher information matrix $\widehat{I}_i^{jj}$, $j = 1, 2, 3, 4$; $i = 1, 2, \dots, 5000$, where $j$ stands for $j$-th elements of parameter vector $\theta = (\alpha, \beta, \mu, \sigma)$;

4.  Compute the average biases (AB), mean squared errors (MSR), coverage probabilities (CP) and average lengths (AW) given by:

$$Bias_{\theta_j}(n) = \frac{1}{5000} \sum_{i=1}^{5000} \left( \widehat{\theta}_{ij} - \theta_j \right),$$

$$MSE_{\theta_j}(n) = \frac{1}{5000} \sum_{i=1}^{5000} (\widehat{\theta}_{ij} - \theta_j)^2,$$

$$CP_{\theta_j}(n) = \frac{1}{5000} 1 \sum_{i=1}^{5000} I \left\{ \widehat{\theta}_{ij} - 1.965.\sqrt{\widehat{I_i^{jj}}} \leq \theta_j \leq \widehat{\theta}_{ij} + 1.965.\sqrt{\widehat{I_i^{jj}}} \right\}$$

and

$$AW_{\theta_j}(n) = \frac{2 \cdot z_{1-\alpha/2}}{5000} \sum_{i=1}^{5000} \sqrt{\widehat{I_i^{jj}}},$$

where $z_{1-\alpha/2}$ is the standardized normal quantile at $100(1 - \alpha)\%$ confidence level and $I\{.\}$ denotes the indicator function.

We repeated these steps based on the sample sizes $n = 100, 110, 120, \ldots, 500$ for the one set of selected values of parameter vector as $(\alpha, \beta, \mu, \sigma) = c(3, 0.5, 0, 1)$. Figures 4–7 show how the AB, MSR, CP and the AW vary with respect to $n$. These results show that the average biases, mean-squared errors and average lengths for each parameter decrease to zero as $n \to \infty$. Additionally, the CP vary with respect to $n$. The associated results of CP corresponds to the nominal coverage probability of 0.95 for two parameters $\beta$ and $\sigma$. The level of CP for the two parameters $\alpha$ and $\mu$ are increasing when $n$ is increased to the level of 0.95.

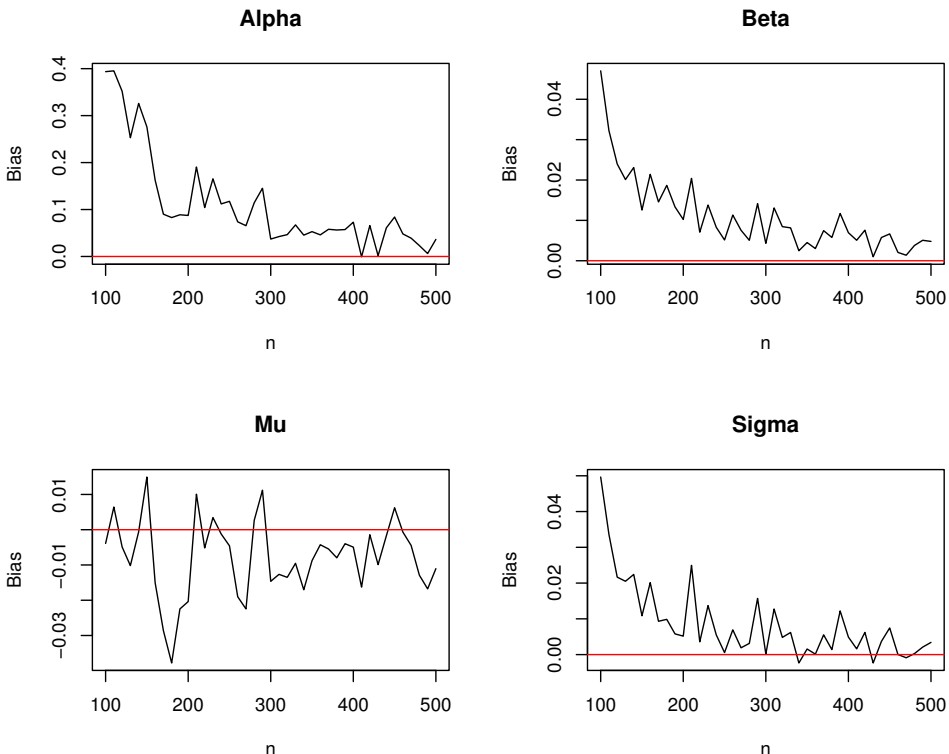

**Figure 4.** AB of the MLE $\widehat{\Theta}$ of the vector parameter $\Theta$.

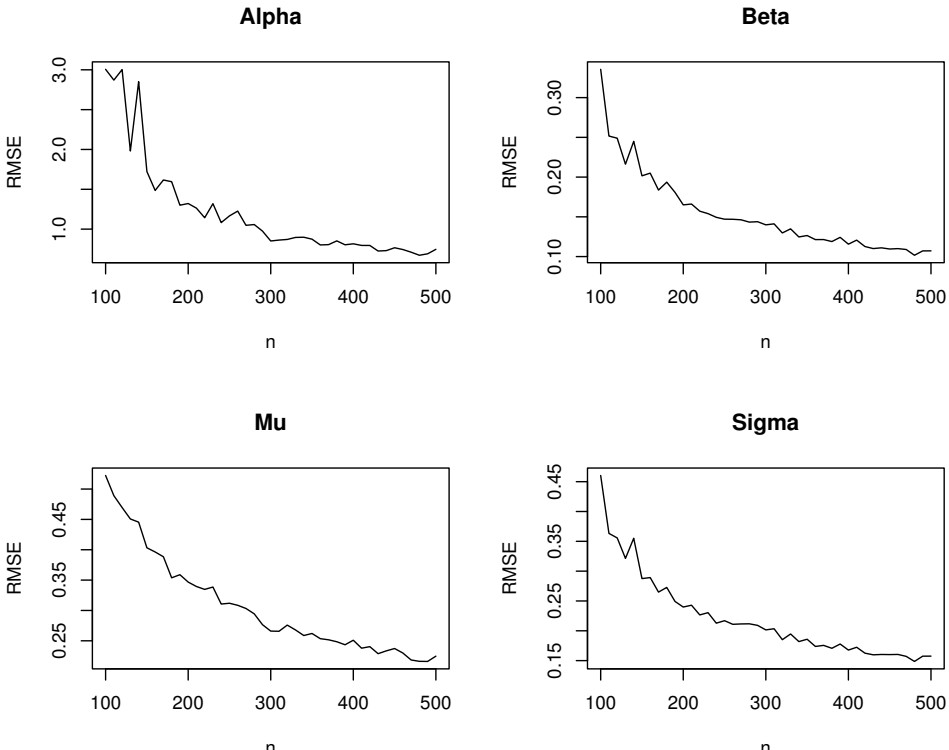

**Figure 5.** RMSE of the MLE $\hat{\Theta}$ of the vector parameter $\Theta$.

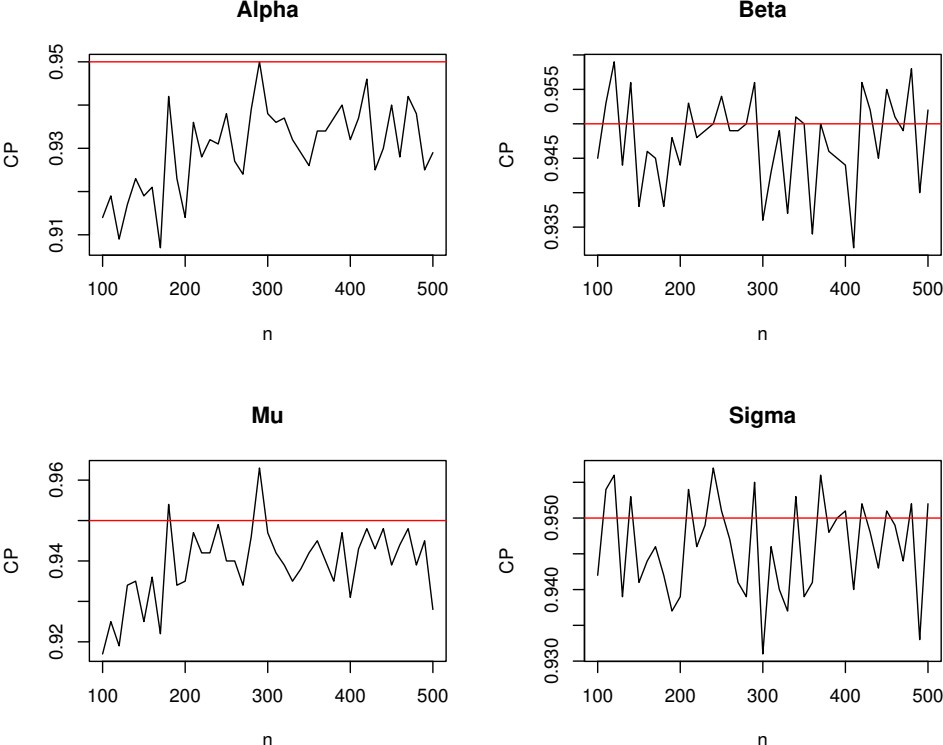

**Figure 6.** CP of 95% confidence intervals of the MLE $\hat{\Theta}$ of the vector parameter $\Theta$.

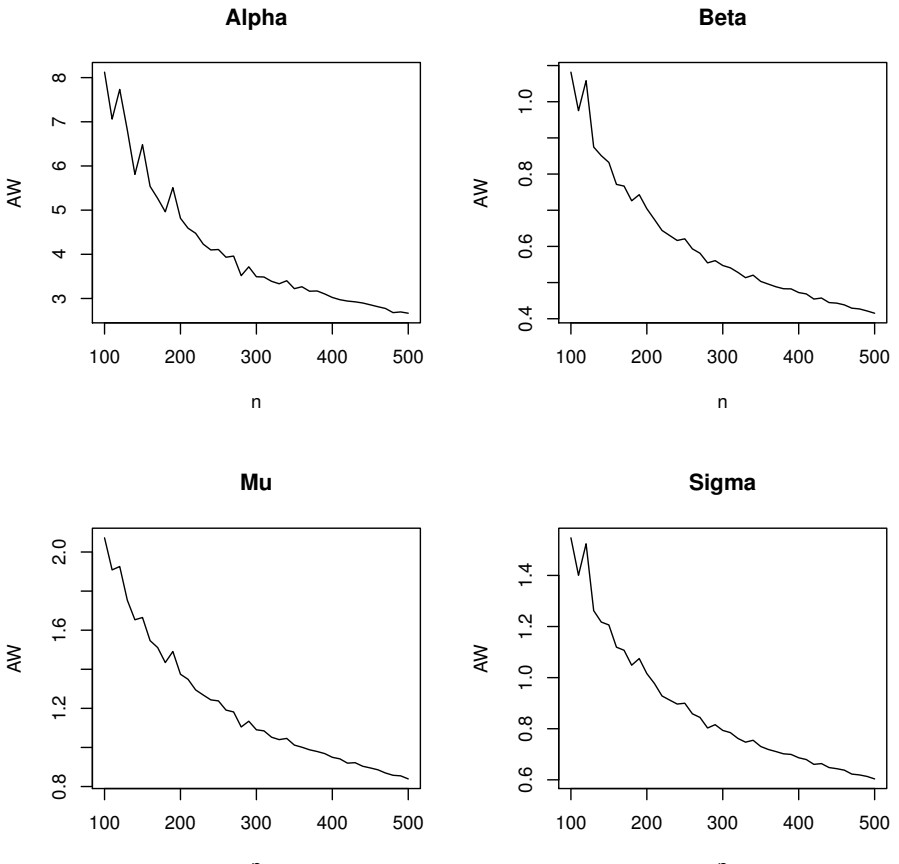

**Figure 7.** AW of 95% confidence intervals of the vector parameter $\Theta$.

## 7. Applications

In this part, we present three applications to investigate the efficiency and flexibility of two sub-classes distributions which formerly defined in Sections 4 and 5. In the first two applications, we present some numerical and graphical results for fitting the special sub-models defined in Section 4. The third application is associated with a survival regression analysis of the ATOLL-W regression model presented in Section 5.

For the first two applications, the goodness-of-fit statistics including the Cramér–von Mises ($W^*$) and Anderson–Darling ($A^*$) test statistics are adopted to compare the fitted models (see [27–29] for more details). The smaller values of $A^*$ and $W^*$ present the better fit to the data. For the sake of comparison, we also consider the Kolmogorov–Smirnov (K-S) statistic and its corresponding *p*-value and the minus log-likelihood function ($-\ell(\psi)$) for the sake of comparison [28,29]. For the third application (covariate censored data), we adopt the AIC and BIC statistics to compare the fitted models since the $A^*$ and $W^*$ statistics are not suitable for censored data.

For the first application, we take the ATOLLN distribution and, for comparison purposes, we fitted the following models to the above datasets:

- The normal distribution;
- The exponentiated normal (EN) distribution;
- The beta normal (BN) distribution [2] with density

$$f_{BN}(x) = \frac{1}{\sigma B(\alpha, \beta)} \left[ \Phi\left(\frac{x-\mu}{\sigma}\right) \right]^{\alpha-1} \left[ 1 - \Phi\left(\frac{x-\mu}{\sigma}\right) \right]^{\beta-1} \phi\left(\frac{x-\mu}{\sigma}\right).$$

- The gamma normal (GN) distribution [30] with density,

$$f_{GN}(x) = \frac{\beta^\alpha}{\sigma\Gamma(a)} \left[ -\log\left\{1 - \Phi\left(\frac{x-\mu}{\sigma}\right)\right\}\right]^{\alpha-1} \left[1 - \Phi\left(\frac{x-\mu}{\sigma}\right)\right]^{\beta-1} \phi\left(\frac{x-\mu}{\sigma}\right).$$

- The Kumaraswamy normal (KN) distribution [3] with density,

$$f_{KN}(x) = \frac{\alpha\,\beta}{\sigma}\left\{\Phi\left[\left(\frac{x-\mu}{\sigma}\right)\right]\right\}^{\alpha-1}\left\{1 - \left[\Phi\left(\frac{x-\mu}{\sigma}\right)\right]^\alpha\right\}^{\beta-1}\phi\left(\frac{x-\mu}{\sigma}\right).$$

- The odd log-logistic normal (OLL-N) distribution (special case of OLLLN distribution when $\beta \to 1$) with density [31],

$$f_{OLL-N}(x) = \frac{\beta\,\phi\left(\frac{x-\mu}{\sigma}\right)[\Phi\left(\frac{x-\mu}{\sigma}\right)]^{\beta-1}[1 - \Phi\left(\frac{x-\mu}{\sigma}\right)]^{\beta-1}}{\sigma\{[1 - \Phi\left(\frac{x-\mu}{\sigma}\right)]^\beta + [\Phi\left(\frac{x-\mu}{\sigma}\right)]^\beta\}^2},$$

where $x \in \mathbb{R}$, $\mu \in \mathbb{R}$, $\alpha > 0$, $\beta > 0$ and $\sigma > 0$.

### 7.1. Failure Times Data

Data 1: First, we analyze the 84 failure times of a particular windshield device. These data were also studied by [32,33].

The MLEs of the parameters, standard errors (SE) (in parentheses) and the goodness-of-fit statistics for failure times data are reported in Table 2. One can see that the ATOLLLN model outperforms all the fitted competitive models under these statistics.

**Table 2.** A summary of model fitting to the failure times data.

| Model | | | | | $-\ell(\psi)$ | $W^*$ | $A^*$ | K-S | $p$-Value |
|---|---|---|---|---|---|---|---|---|---|
| ATOLLN$(\mu,\sigma,\alpha,\beta)$ | 2.903 | 0.495 | −1.285 | 0.319 | **126.077** | **0.031** | **0.312** | **0.05** | **0.983** |
| | (0.200) | (0.162) | (0.630) | (0.161) | | | | | |
| OLL-N$(\mu,\sigma,\beta)$ | 2.626 | 0.602 | 0.452 | | 127.062 | 0.075 | 0.523 | 0.095 | 0.407 |
| | (0.126) | (0.218) | (0.232) | | | | | | |
| ATN$(\mu,\sigma,\alpha)$ | 2.615 | 1.121 | 0.467 | | 128.111 | 0.087 | 0.585 | 0.089 | 0.510 |
| | (0.476) | (0.114) | (2.011) | | | | | | |
| Normal$(\mu,\sigma)$ | 2.557 | 1.112 | | | 128.119 | 0.091 | 0.607 | 0.092 | 0.444 |
| | (0.121) | (0.086) | | | | | | | |
| EN$(\mu,\sigma,\alpha)$ | 1.823 | 1.339 | 1.954 | | 128.064 | 0.074 | 0.521 | 0.084 | 0.560 |
| | (2.342) | (0.701) | (3.864) | | | | | | |
| BN$(\mu,\sigma,\alpha,\beta)$ | 0.808 | 2.443 | 7.113 | 2.469 | 128.085 | 0.074 | 0.519 | 0.084 | 0.562 |
| | (7.144) | (8.149) | (48.513) | (14.595) | | | | | |
| GaN$(\mu,\sigma,\alpha,\beta)$ | 2.805 | 0.541 | 0.290 | 0.197 | 127.757 | 0.057 | 0.438 | 0.074 | 0.710 |
| | (1.057) | (0.264) | (0.381) | (0.215) | | | | | |
| KwN$(\mu,\sigma,\alpha,\beta)$ | 1.653 | 0.747 | 0.918 | 0.319 | 127.848 | 0.063 | 0.468 | 0.079 | 0.641 |
| | (1.063) | (0.534) | (1.013) | (0.518) | | | | | |
| ATOLLW$(\lambda,\gamma,\alpha,\beta)$ | 0.288 | 7.080 | 1.993 | 0.341 | 126.95 | 0.088 | 0.672 | 0.067 | 0.84 |
| | (0.011) | (0.034) | (0.603) | (0.036) | | | | | |

The fitted densities and histogram of the data are displayed in Figure 8. For failure times, we note that the fitted ATOLLN distribution best captures the empirical histogram.

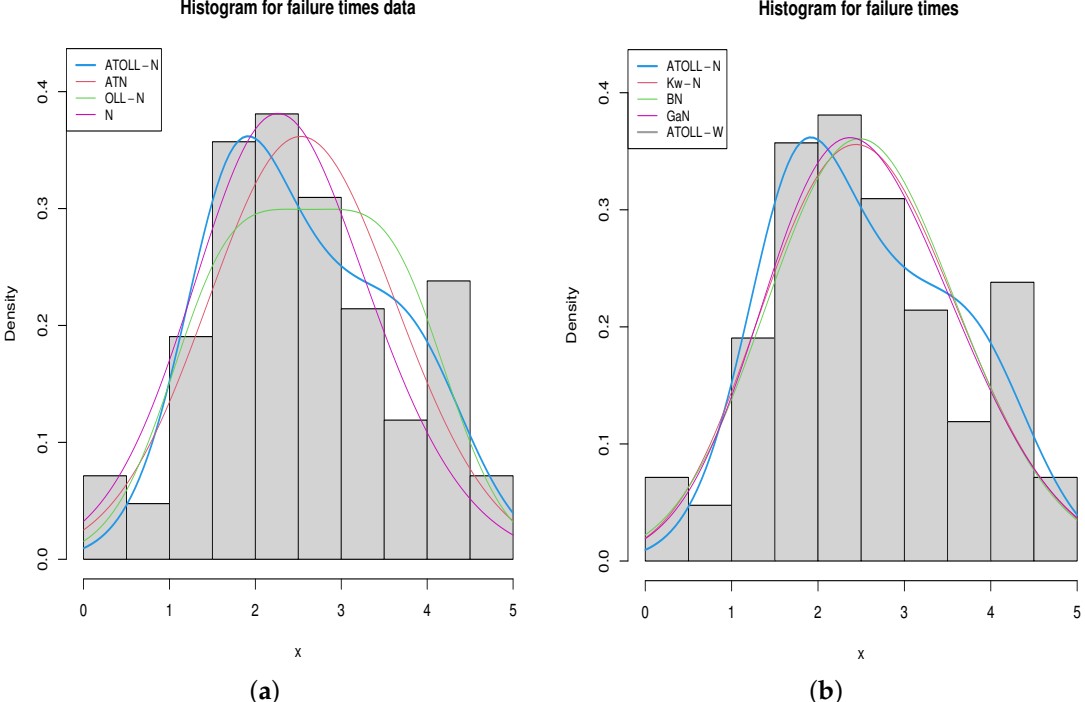

(**a**)　　　　　　　　　　　　(**b**)

**Figure 8.** Histogram and density plots for failure times data. Plots for (**a**) sub-models and (**b**) others models.

## 7.2. Windshield Device Data

Data 2: Second, we examine 23 failure times of a particular windshield device. These data were also studied by [32,33]. The data, referred as windshield device, are: 2.160, 0.746, 0.402, 0.954, 0.491, 6.560, 4.992, 0.347, 0.150, 0.358, 0.101, 1.359, 3.465, 1.060, 0.614, 1.921 4.082, 0.199, 0.605 0.273 0.070 0.062 5.320.

Here, we fit the ATOLLW distribution and some of its sub-models, odd log-logistic Weibull (OLL-W), beta Weibull (BW), Kumaraswamy–Weibull (kwW), gamma Weibull (GaW) and exponentiated Weibull (EW) distributions to the windshield device data. Similar numerical results are provided in Table 3 for windshield device data as well as data failure times data. It is immediately seen that the ATOLLW model outperforms all the fitted competitive models under the model selection criteria presented for the first data application.

**Table 3.** A summary of model fitting to the windshield device data.

| Model | | | | | $-\ell(\psi)$ | $W^*$ | $A^*$ | K-S | *p*-Value |
|---|---|---|---|---|---|---|---|---|---|
| ATOLLW$(\lambda, \gamma, \alpha, \beta)$ | 0.241 (0.026) | 6.068 (0.667) | 5.210 (2.590) | 0.156 (0.033) | **29.95** | **0.023** | **0.174** | **0.071** | **0.999** |
| OLL-W$(\lambda, \gamma, \beta)$ | 0.685 (0.302) | 0.636 (0.665) | 1.315 (1.524) | | 32.47 | 0.052 | 0.368 | 0.102 | 0.948 |
| ATW$(\lambda, \gamma, \alpha)$ | 0.369 (0.375) | 0.909 (0.204) | 2.500 (4.156) | | 32.27 | 0.043 | 0.318 | 0.108 | 0.924 |
| Weibull$(\lambda, \gamma)$ | 0.718 (0.196) | 0.807 (0.129) | | | 32.51 | 0.065 | 0.431 | 0.118 | 0.866 |
| EW$(\lambda, \gamma, \alpha)$ | 41.358 (277.983) | 0.298 (0.279) | 10.443 (30.099) | | 31.83 | 0.023 | 0.211 | 0.096 | 0.967 |
| kwW$(\lambda, \gamma, \alpha, \beta)$ | 888.601 (0.017) | 0.414 (0.001) | 281.499 (0.035) | 0.088 (0.018) | 30.97 | 0.024 | 0.226 | 0.103 | 0.945 |
| BW$(\lambda, \gamma, \alpha, \beta)$ | 41.899 (0.084) | 0.625 (0.020) | 5.919 (3.737) | 0.104 (0.025) | 31.35 | 0.021 | 0.192 | 0.097 | 0.964 |
| GaW$(\lambda, \gamma, \alpha, \beta)$ | 452.763 (0.240) | 0.449 (0.002) | 5.504 (1.684) | 2.909 (0.814) | 31.67 | 0.041 | 0.297 | 0.101 | 0.952 |
| ATOLLN$(\mu, \sigma, \alpha, \beta)$ | 3.051 (0.003) | 0.435 (0.002) | 4.636 (0.201) | 0.103 (0.270) | 38.34 | 0.190 | 1.117 | 0.201 | 0.270 |

The fitted densities and histogram of the windshield device data are displayed in Figure 9. This figure shows that the fitted ATOLLW distribution best captures the empirical histogram among the considered competitor models.

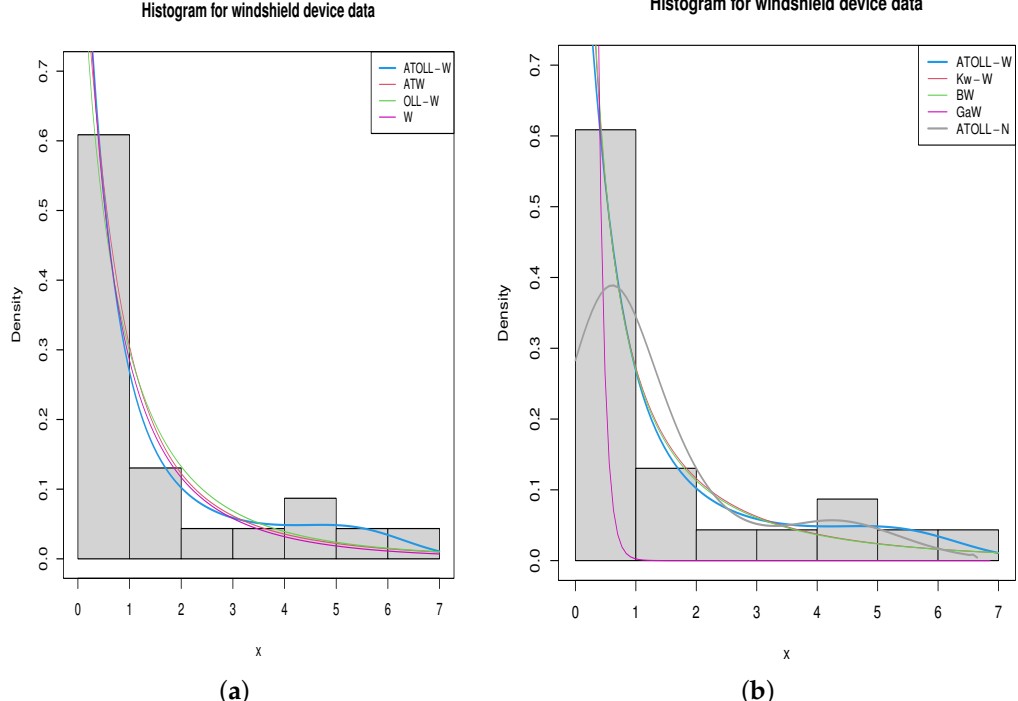

**Figure 9.** Histogram and density plots for windshield device data. Plots for (**a**) sub-models and (**b**) others models.

We note that the ATOLLN and ATOLLW models outperform all the fitted competitive models under the selected criterion for the datasets' failure times and windshield device, respectively.

### 7.3. Third Application: Regression Analysis

Survival regression analysis has been developed in several forms. One of them is the non-parameteric, where Kaplan–Meier estimation [34] is highlighted. The Kaplan–Meier estimate is a common way of obtaining the survival curve using probabilities of an event's occurrence at a time. In this section, we provided a parametric approach as a counterpart, where we fit the LATOLLW regression to the heart transplant data. The current data are available in a `survival` package of `R` software. The considered survival regression model based on response variable $y_i$ and covariate variables $(v_{i1}, v_{i2}, v_{i3})$ is formulated as:

$$y_i = \tau_0 + \tau_1 v_{i1} + \tau_2 v_{i2} + \tau_2 v_{i3} + \sigma z_i, \ i = 1, ..., n,$$

where $y_i$ is distributed as the LATOLLW distribution and the covariate random variables are described as:

- $v_{i1} = $ age;
- $v_{i2} = $ previous surgery (0 = no; 1 = yes);
- $v_{i3} = $ transplant (0 = no; 1 = yes).

#### 7.3.1. Parameter Estimation

A summary of model fitting based on MLE discussion for the heart transplant data is provided in Table 4. We fit the LATOLLW regression model to this dataset and compare the results with LBXII-W, LOLLW and log-Weibull distributions. For more details about these competitor models, see [23]. We also consider another alternative models such as

a log-log mean Weibull (LLMW) regression proposed by [35] and log exponential-Pareto (LEP) regression model proposed by [36]. The estimated parameters, standard errors (given in parentheses) and AIC and BIC measures as well as corresponding p-values in [.] are reported in Table 4. We conclude that the estimated regression parameters are statistically significant at the 5% level.

**Table 4.** A summary of fitted regression models to the heart transplant data.

| Model | $\tau_0$ | $\tau_1$ | $\tau_2$ | $\tau_3$ | $\sigma$ | $\alpha$ | $\beta$ |
|---|---|---|---|---|---|---|---|
| Arctan-Weibull | 9.3779 | −0.0689 | 1.3418 | 2.2273 | 0.2034 | 12.8680 | 0.1687 |
| | (0.2128) | (0.0054) | (0.1227) | (0.2131) | (0.0053) | (4.9179) | (0.0171) |
| | [0.0072] | [0.0247] | [0.0290] | [0.0304] | | | |
| | **AIC = 330.7971** | **BIC = 349.2402** | | | | | |
| LBXII-W | 4.519 | −0.055 | 1.747 | 2.571 | 2.638 | 3.666 | 0.175 |
| | (1.053) | (0.019) | (0.546) | (0.359) | (1.142) | (1.616) | (0.098) |
| | [<0.001] | [0.004] | [0.001] | [<0.001] | | | |
| | $AIC = 343.3$ | $BIC = 361.8$ | | | | | |
| LOLLW | 8.744 | −0.076 | 1.405 | 2.591 | 6.203 | 4.628 | |
| | (1.760) | (0.019) | (0.574) | (0.388) | (4.685) | (3.530) | |
| | [<0.001] | [<0.001] | [0.016] | [<0.001] | | | |
| | $AIC = 347.5$ | $BIC = 363.4$ | | | | | |
| log-Weibull | 7.972 | −0.092 | 1.214 | 2.537 | 1.465 | | |
| | (0.934) | (0.020) | (0.647) | (0.373) | (0.131) | | |
| | [<0.001] | [<0.001] | [0.063] | [<0.001] | | | |
| | $AIC = 353.4$ | $BIC = 366.6$ | | | | | |
| LLMW | 6.617 | −0.091 | 1.640 | 2.591 | 2.618 | 1.169 | 0.013 |
| | (1.122) | (0.025) | (0.624) | (0.402) | (0.151) | (0.057) | |
| | [0.048] | [0.036] | [0.036] | [0.046] | | | |
| | $AIC = 349.6$ | $BIC = 365.5$ | | | | | |
| LEP | 5.1321 | −0.0923 | 1.214127 | 2.537713 | 1.4655 | | 0.1439 |
| | (11.3276) | (0.0206) | (0.6469) | (0.3733) | (0.1314) | | (1.1088) |
| | [0.6505] | [<0.001] | [<0.061] | [<0.001] | | | |
| | $AIC = 355.42$ | $BIC = 371.22$ | | | | | |

### 7.3.2. Results of Residual Analysis

The suitability of the fitted LATOLLW regression model is evaluated by residual analysis. The plot of the modified deviance residuals is displayed in Figure 10, which reveals that the fitted LATOLLW regression provides a good fit to the current data.

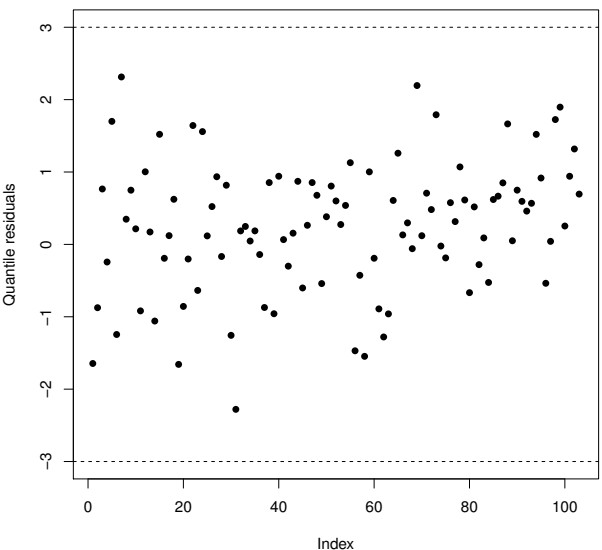

**Figure 10.** Index plot of the quantile residuals.

*7.4. Bayesian Regression Analysis: Heart Transplant Data*

From (24), we can see that there is no explicit form for the Bayesian estimators under the loss functions considered in Table 1, so we use Gibbs sampling technique and MCMC procedure based on 10,000 replicates to obtain Bayesian estimators for the heart transplant data. A summary of Bayesian analyses (point and interval estimations with related posterior risk) are reported in Tables 5 and 6. Table 6 provides 95% credible and HPD intervals for each parameter of the LATOLLW distribution. Moreover, we provide the posterior summary plots in Figures 11–13. These plots confirm that the convergence of sampling process occurred.

**Table 5.** A summary of the Bayesian analysis of the LATOLLW regression based on heart transplant data.

| Loss Function | $\tau_0$ | $\tau_1$ | $\tau_2$ | $\tau_3$ | $\sigma$ | $\alpha$ | $\beta$ |
|---|---|---|---|---|---|---|---|
| SELF | 4.7244 | −0.0122 | 1.0151 | 1.6356 | 1.1790 | 0.6499 | 0.8541 |
| | (0.4974) | (0.0004) | (0.4019) | (0.1110) | (0.0169) | (0.5521) | (0.0164) |
| WSELF | 4.5712 | −0.0141 | 0.7837 | 1.5591 | 1.1658 | 0.0818 | 0.8357 |
| | (0.1532) | (0.0019) | (0.2314) | (0.0766) | (0.0132) | (0.5680) | (0.0185) |
| MSELF | 4.3337 | −0.0013 | 0.00289 | 1.4721 | 1.1537 | 0.0015 | 0.8174 |
| | (0.0519) | (0.9999) | (0.9963) | (0.0557) | (0.0104) | (0.9807) | (0.0219) |
| PLF | 4.7768 | 0.0235 | 1.1968 | 1.6692 | 1.1862 | 0.9872 | 0.8637 |
| | (0.1047) | (0.0715) | (0.3633) | (0.0672) | (0.0143) | (0.6745) | (0.0191) |
| KLF | 4.6472 | 0.0131 | 0.8919 | 1.5969 | 1.1724 | 0.2307 | 0.8449 |
| | (0.0332) | (0.1427) | (0.2762) | (0.0485) | (0.0113) | (3.6347) | (0.0220) |
| LINEXLF (c = 1) | 4.3559 | −0.0124 | 0.8571 | 1.5794 | 1.1709 | 0.4785 | 0.8462 |
| | (0.3684) | (0.0002) | (0.1580) | (0.0562) | (0.0081) | (0.1713) | (0.0079) |
| LINEXLF (c = −1) | 4.9239 | −0.0120 | 1.3049 | 1.6903 | 1.1879 | 1.1870 | 0.8627 |
| | (0.1995) | (0.0002) | (0.2898) | (0.0546) | (0.0089) | (0.5372) | (0.0085) |
| GELF (c = 1) | 4.5712 | −0.0141 | 0.7838 | 1.5590 | 1.1658 | 0.0818 | 0.8357 |
| | (0.0189) | (0.189) | (0.0159) | (0.0253) | (0.0055) | (1.4457) | (0.0109) |
| GELF (c = −1) | 4.7244 | −0.0122 | 1.0151 | 1.6356 | 1.1790 | 0.6499 | 0.8541 |
| | (0.0140) | (0.1201) | (0.3140) | (0.0225) | (0.0057) | (0.6258) | (0.0109) |

**Table 6.** Credible and HPD intervals of the parameters $\tau_0$, $\tau_1$, $\tau_2$, $\tau_3$, $\sigma$, $\alpha$ and $\beta$ for heart transplant data.

| | Credible Interval | HPD Interval |
|---|---|---|
| $\tau_0$ | (4.4197, 5.1520) | (3.5680, 6.3040) |
| $\tau_1$ | (−0.0237, −0.0049) | (−0.0523, 0.0186) |
| $\tau_2$ | (0.6154, 1.2680) | (−0.0832, 2.3680) |
| $\tau_3$ | (1.4050, 1.8661) | (0.9674, 2.2480) |
| $\sigma$ | (1.0900, 1.2470) | (0.9403, 1.4330) |
| $\alpha$ | (0.1778, 0.8085) | (0.0004, 2.2740) |
| $\beta$ | (0.7799, 0.9169) | (0.6169, 1.1350) |

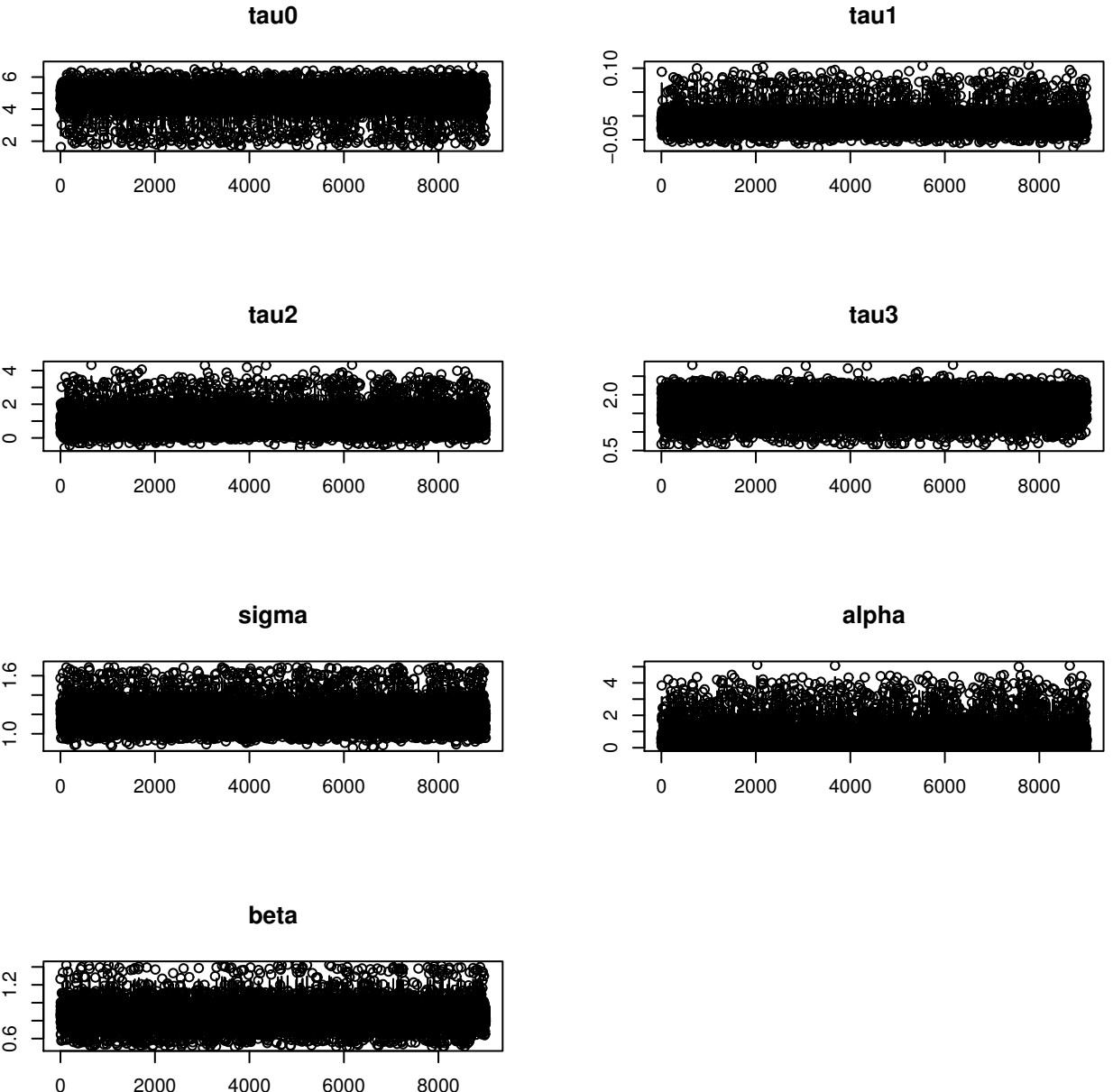

**Figure 11.** Trace plots of Bayesian analysis and performance of Gibbs sampling for the each parameter of LATOLLW distribution based on the heart transplant data.

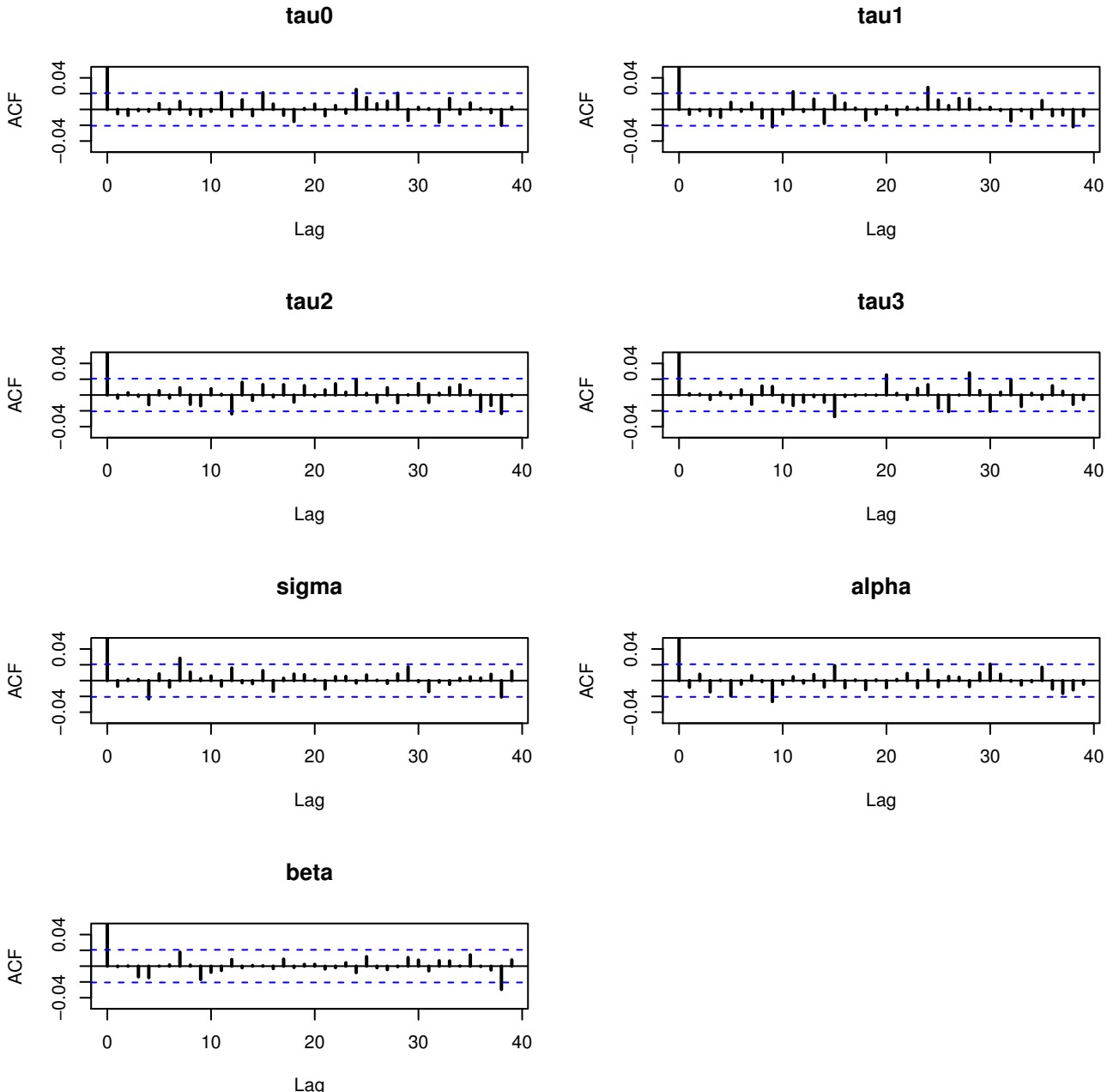

**Figure 12.** Autocorrelation race plots of Bayesian analysis and performance of Gibbs sampling for the each parameter of LATOLLW distribution based on the heart transplant data.

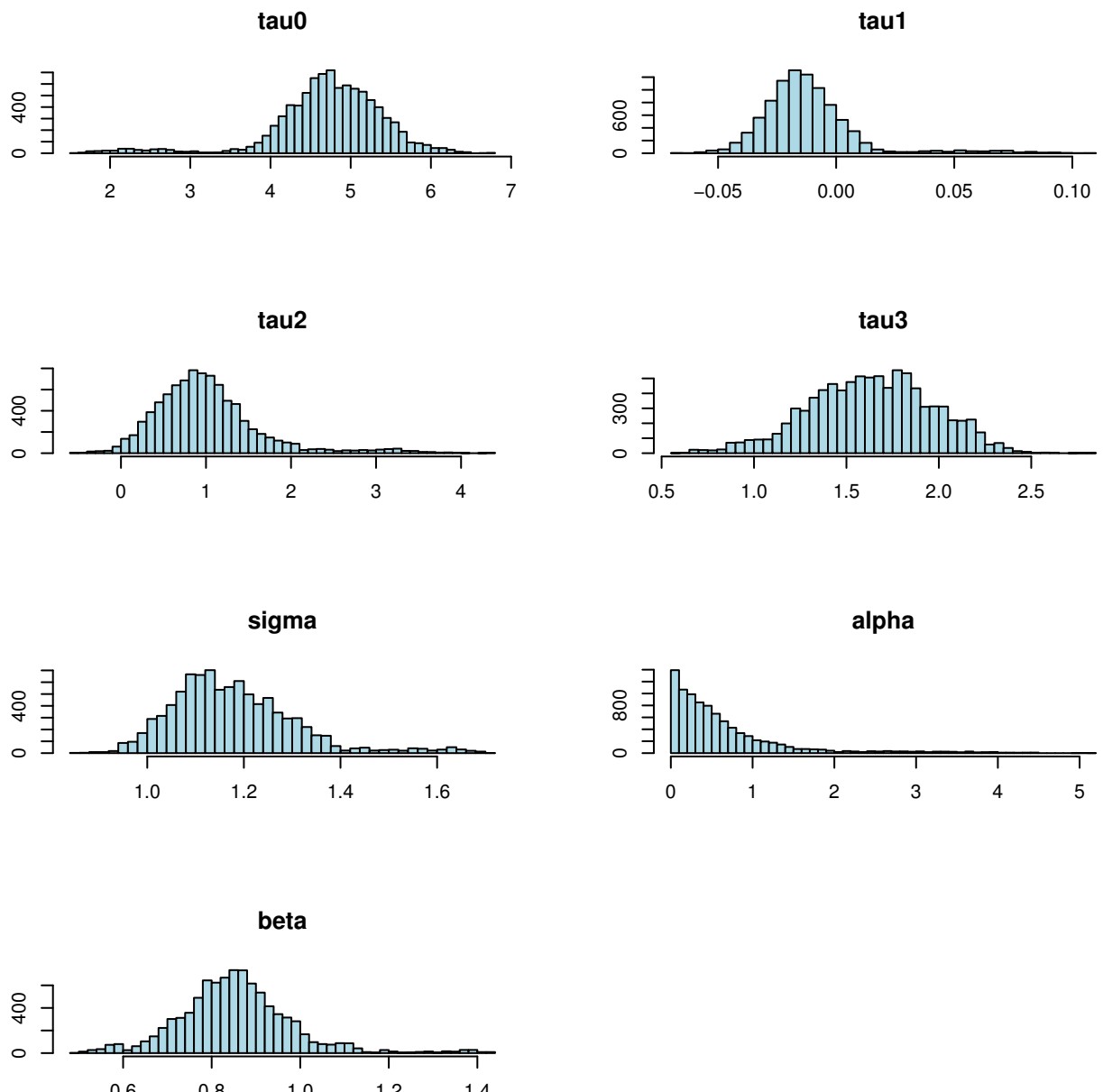

**Figure 13.** Histogram plots of Bayesian analysis and performance of Gibbs sampling for the each parameter of LATOLLW distribution based on the Heart transplant data.

## 8. Conclusions

A new class of lifetime distributions was introduced via compounding odd log logistic distribution and the arctan function. Two special sub-models of this class were proposed by considering the Weibull and normal distributions instead of the baseline distribution. We have also provided a survival regression model based on Weibull distribution and a comprehensive discussion about Bayesian inference for the parameters of this survival regression model were studied under various loss functions. Numerical analyses of fitting two univariate real datasets were provided via a maximum likelihood approach and the corresponding plots were drawn to evaluate these results visually. The data analysis empirically proved that the proposed distributions provide a better fit than their competing distributions. Finally, the performance of the survival regression sub-model was examined in terms of maximum likelihood and Bayesian procedures for a real example of observations with covariate variables.

**Author Contributions:** Conceptualization, O.K. and M.A.; methodology, O.K. and M.A.; software, O.K. and M.A.; validation, O.K., M.A. and J.E.C.-R.; formal analysis, O.K.; investigation, O.K., M.A., J.E.C.-R. and H.H.; resources, J.E.C.-R.; data curation, O.K. and M.A.; writing—original draft preparation, O.K., M.A. and J.E.C.-R.; writing—review and editing, O.K., M.A., J.E.C.-R. and H.H.; visualization, O.K. and M.A.; supervision, J.E.C.-R. and H.H.; project administration, O.K.; funding acquisition, J.E.C.-R. All authors have read and agreed to the published version of the manuscript.

**Funding:** This research was fully supported by FONDECYT (Chile) grant No. 11190116.

**Institutional Review Board Statement:** Not applicable.

**Informed Consent Statement:** Not applicable.

**Data Availability Statement:** Data provided in the paper.

**Acknowledgments:** The authors thank the editor and three anonymous referees for their helpful comments and suggestions.

**Conflicts of Interest:** The authors declare that there is no conflict of interest in the publication of this paper.

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
