# Peer review of "Arctan-Based Family of Distributions: Properties, Survival Regression, Bayesian Analysis and Applications"

_axioms, doi:10.3390/axioms11080399_

Round 1

Reviewer 1 Report

The manuscript is well-written and constitutes a good contribution to the field of Statistical Distributions and Applications. I advise some minor revision.

1) please correct the  parentheses height in equation (13)

2) It would be nice to have some plots of the PDFs of the two sub-models mentioned in section 4.

3) Figures 2, 3, and 4 evidence high volatility as a function of the sample size n. This is possible due to a high sampling error. To achieve results closer to the unknown true value, I advise the author to repeat the simulation study with more samples. For example, use 5000 samples, instead of 1000 samples.

4) Page 9: Why was the value 1.965 used to compute the CP, instead of the standardized normal quantile at 97.5%. 

Also, since the CP is usually smaller than 95%, I advise the use of the quantile of a Student's t with n degrees of freedom, instead of the quantile of the standardized normal distribution. This has no theoretical justification.

5) Tables 2 and 3: Most software computes the corresponding p-value of the Kolmogorov-Smirnov assuming the parameters are known (not estimated from the sample). When the model parameters are estimated from the sample, the p-value is computed differently. I advise the authors to clarify how was computed the p-value.

Author Response

Please see attached letter.

Reviewer 2 Report

1. The article lacks criteria for checking different types of distribution laws.

2. It is not clear what the regression equation looks like and its practical application.

3. It is necessary to provide more practical examples with different types of distribution laws, adding them to the example of heart transplantation.

4. When considering survival regression, I propose in the introductory section to analyze relevant works on nonparametric survival (at least Kaplan-Meier estimation). Also, when considering the examples, one should mention the non-parametric survival and compare it with the parametric one.

Author Response

Please see attached letter.

Reviewer 3 Report

My comments file is attached.

Author Response

Please see attached letter.
